# What Drives Caterpillar Guilds on a Tree: Enemy Pressure, Leaf or Tree Growth, Genetic Traits, or Phylogenetic Neighbourhood?

**DOI:** 10.3390/insects13040367

**Published:** 2022-04-08

**Authors:** Freerk Molleman, Urszula Walczak, Iwona Melosik, Edward Baraniak, Łukasz Piosik, Andreas Prinzing

**Affiliations:** 1Department of Systematic Zoology, Faculty of Biology, Institute of Environmental Biology, Adam Mickiewicz University in Poznań, Uniwersytetu Poznańskiego 6, 61-614 Poznań, Poland; urszula.walczak@amu.edu.pl (U.W.); edward.baraniak@amu.edu.pl (E.B.); 2Department of Genetics, Faculty of Biology, Institute of Experimental Biology, Adam Mickiewicz University in Poznań, Uniwersytetu Poznańskiego 6, 61-614 Poznań, Poland; iwona.melosik@amu.edu.pl; 3Department of General Botany, Faculty of Biology, Institute of Experimental Biology, Adam Mickiewicz University in Poznań, Uniwersytetu Poznańskiego 6, 61-614 Poznań, Poland; 4Ecosystèmes Biodiversité Evolution (ECOBIO), Campus de Beaulieu, Université de Rennes 1, 35042 Rennes, France; andreas.prinzing@univ-rennes1.fr

**Keywords:** budburst phenology, population genetics, dispersal limitation, genome size, phylogenetic isolation, leaf miner, shelter building, concentration effect, functional traits

## Abstract

**Simple Summary:**

The number of insects that feed on an individual tree can be influenced by the growth of leaves during the season, the size of the tree, genetic traits that affect leaf quality, and by the tree species that surround it. To estimate the relative importance of these processes, we determined the date on which leaves start unfolding in spring, trunk diameter, genotype, and neighbourhood of sessile oak trees, and sampled their caterpillar communities. We found that free-living caterpillars were less abundant on older leaves. Caterpillars were less diverse and experienced higher parasitism on larger trees. Leaf-mining casebearers were more abundant on trees that were genetically more homozygous. However, genome size was not important for any guild. In contrast to most previous studies, oaks surrounded by distantly related tree species tended to have higher caterpillar densities. Neighbourhoods were also related to species composition and diversity, but not to the average wingspans or specialization of species. Common species were less abundant on trees with high parasitism rates. Our results suggest that trees are not always better off in diverse forests, as large trees surrounded by distantly related species might actually suffer more insect damage.

**Abstract:**

Communities of herbivorous insects on individual host trees may be driven by processes ranging from ongoing development via recent microevolution to ancient phylogeny, but the relative importance of these processes and whether they operate via trophic interactions or herbivore movement remains unknown. We determined the leaf phenology, trunk diameter, genotype, and neighbourhood of sessile oak trees (*Quercus petraea*), and sampled their caterpillar communities. We found that leaf development across a time period of days related to free-living caterpillars, which disappeared with leaf age. Tree growth across decades is related to increased parasitism rate and diversity of herbivores. The microevolution of oak trees across millennia is related to the abundance of leaf-mining casebearers, which is higher on more homozygous oaks. However, oak genome size was not important for any guild. In contrast to most previous studies, the phylogenetic distance of oaks from their neighbours measured in millions of years was associated with higher abundances of entire caterpillar guilds. Furthermore, on trees surrounded by only distantly related tree species, parasitism tended to be lower. Lower parasitism, in turn, was associated with higher abundances of codominant caterpillar species. Neighbourhoods and traits of trees were also related to community composition and diversity, but not to the average wingspans or specialization of species, consistent with the assembly of herbivore communities being driven by leaf traits and parasitism pressure on trees rather than by insect movement among trees. However, movement in rarer species may be responsible for concentration effects in more phylogenetically distant neighbourhoods. Overall, we suggest that the assembly of insects on a tree is mostly driven by trophic interactions controlled by a mosaic of processes playing out over very different time scales. Comparisons with the literature further suggest that, for oak trees, the consequences of growing amongst distantly related tree species may depend on factors such as geographic region and tree age.

## 1. Introduction

When an insect is searching for tree leaves in a forest, it faces leaf characteristics that result from processes that play out across days to millions of years. On the shortest time scale of days to weeks, leaves burst from buds and mature, and on the scale of decades, trees mature and grow larger. On the scale of centuries to millennia, genetic lineages within the host species become isolated, evolve, disperse, and outbreed. Finally, the tree species surrounding a tree have diverged from the host species during millions of years of evolution. These different processes have been shown to affect insect abundance or diversity on individual trees (introduced below), but their relative importance remains unknown.

As leaves of a host tree develop, their quality for herbivores decreases [1,2], reducing the abundance and diversity of herbivorous insects within a few weeks [3,4]. Trees may thus differ in insect communities at a given time due to variation in the timing of budburst ([5,6,7,8,9,10,11]; Figure 1, arrow 1). Furthermore, populations of hosts consist of individuals with different microevolutionary backgrounds, causing differences in palatability and defence traits [12,13,14]. Genetic lineages within plant species may hence harbour distinct insect lineages ([10,15,16,17,18,19,20,21]; Figure 1, arrow 2), but see [22,23,24]. Individuals will further differ in levels of heterozygosity, depending on the degree to which they are the results of crosses between lineages. The heterozygosity of a tree can increase the within-tree diversity of expressed traits. This within individual genetic diversity, can benefit herbivores (as was demonstrated using interspecific crosses, [25]), but can also suppress them due to the expression of a wider variety of defence traits ([26,27]; Figure 1, arrow 2). Individuals within a plant population can also vary in genome size [28,29,30]. While genome size affects insect herbivores across species [31], whether intraspecific variation in genome size of plants affects insect communities has rarely been studied in the field ([32]; Figure 1, arrow 2). Finally, the neighbouring trees may also affect leaf quality through resource competition [33,34], cooperation in the rhizosphere [35], or allelopathic interference [36], or they may reflect environmental conditions that affect leaf quality [37].

Besides leaf traits, herbivore communities on trees can also be affected by insect movements, which may be driven by tree neighbourhood and tree size. When neighbouring trees are of the same or closely related species, they can mutually act as sources of specialized herbivorous insects [38,39], and thus increase the rate of movement of specialized herbivores onto the individual tree. In contrast, trees surrounded by distantly related neighbours may experience lower levels of immigration by such specialists ([40]; Figure 1, arrow 4). Phylogenetic isolation (the average evolutionary distance to neighbouring trees in millions of years) can dramatically reduce leaf damage and herbivore abundance on trees [40,41,42]. Furthermore, a larger and thus older tree might have had more time to accumulate herbivore species, similar to older islands having accumulated more species [43]. Larger trees also form larger resource patches that are easier to detect (plant apparency hypothesis; [44,45]), and where the chance of local extinction is lower [43]. Species accumulation, apparency, and extinction risk may partly explain why herbivorous insect communities are often more species-rich on larger trees ([45,46,47]; Figure 1, arrow 5). These processes of insect colonization and extinction on individual trees can be independent of insect species traits and species identity (neutral), and thus only affect the diversity and abundance of insects, not community composition. In contrast, due to possible interactions between the neighbourhood and insect traits, we would expect the neighbourhood to also affect insect community composition [40,48]. In particular, we would expect a higher proportion of good dispersers and host plant generalists on trees with more distantly related neighbours [40].

The population density of insects feeding on a tree will be affected by natural enemy pressure, which can also vary among individual trees. Because natural enemy pressure affects herbivore performance, it can be regarded as one of the characteristics of a host tree, similar to leaf quality (Figure 1, arrow 6). Natural enemy pressure may vary among individual trees due to variation in the emission of herbivore-induced plant volatiles that natural enemies may use to orient toward herbivore-infested trees [49]. Enemy pressure on insects may also decline when host trees are surrounded by distantly related neighbours due to isolation and odor masking ([50,51,52]; Figure 1, arrow 7). Furthermore, specialized natural enemies such as parasitoids might be less at risk of local extinction on larger trees ([53]; Figure 1, arrow 8). Enemy pressure could also increase herbivore diversity by reducing the competitive dominance of some herbivore species over others [54]. Enemy pressure should hence be part of any comprehensive model explaining the abundance, diversity, and community composition of insect herbivores on trees [55,56]. Inversely, processes playing out over different time scales need to be accounted for when studying natural enemy pressure (Figure 1, arrows 7 and 8).

The degree of phylogenetic isolation of a tree may modify the effect of leaf traits on insect communities by affecting local adaptation within insect species. Herbivorous insects can show adaptations to individual trees or lineages of trees within a forest [57,58,59,60,61,62,63]. The degree of local adaptation is likely affected by insect movements and subsequent gene flow. On the one hand, more colonizers provide more genetic variation for natural selection to act on [64,65]. On the other hand, local adaptations may be swamped by a high rate of immigration of non-adapted new colonizers [63,66]. Such swamping may be limited enough for local adaptation to occur even when distances between conspecific trees are small compared to maximum dispersal distances of insects, as realized dispersal distances are often much shorter [67,68]. Furthermore, the rate of immigration can be especially low for species with wingless females [69] or when females mainly oviposit near their eclosion site [62,70]. Therefore, phylogenetically isolated trees might be sites of more (less swamping) or less (less genetic variation for selection to act on) local adaptation than less isolated trees. The degree of local adaptation will be reflected in insect abundances, as it would allow insects to thrive on trees with unfavourable traits, and hence weaken the relationship between tree traits and insect abundance (Figure 1, arrow 9 representing the interaction term phylogenetic isolation: trait).

How these characteristics of a given tree affect insects may also be contingent on their guild. Guilds of insect herbivores vary in their ability to manipulate leaves, affecting how they respond to leaf quality. Among caterpillars, leaf miners are particularly adept at manipulating leaf quality [71], and many species of leaf miner avoid structural and/or chemical defences that tend to concentrate in the cuticle and epidermis by feeding in the nutrient-rich mesophyll [72,73]. Therefore, natural leaf quality may be less relevant to leaf miners than to ectophages. Guilds also vary in the degree of host-plant specialization, and thus in the number of neighbouring tree species they can feed on. In particular, leaf miners tend to be highly host-plant-specific [74,75], and semi-concealed caterpillars tend to be more host-plant-specific than free-living ones [38]. Furthermore, herbivorous insect species will overall be more abundant than their parasitoids. Therefore, herbivorous insects might have a lower risk of local stochastic extinction on a small, isolated resource patch than their parasitoids. Finally, leaf miners and semi-concealed caterpillars can be sheltered against predation [76,77,78,79,80]. A lower mortality cost of slow development may permit these caterpillars to feed for a longer time, and thus to feed on leaves that are harder to digest. Different guilds need to be studied simultaneously to determine whether an insects’ biology influences the various processes determining abundance and community composition on individual trees.

While identifying the importance of any single one of these processes alone may still be a challenge, it will say little about the relative importance of each process compared to others. It might even produce a pseudo-correlation driven by processes not accounted for (e.g., [81,82] showing that the phylogenetic distance of neighbourhoods drives leaf development of oaks). Therefore, identifying the absolute and relative role of a given process at one scale requires accounting for all processes at all scales simultaneously. In addition, we predict that local adaptation causes statistical interactions between phylogenetic isolation and characteristics that affect leaf quality (Figure 1, arrow 9). Finally, differentiation between guilds has rarely been done (but see, e.g., [83]) and, to our knowledge, never across multiple processes.

To gauge the relative contribution of the various processes discussed above (Figure 1), we sampled and reared spring caterpillars from sessile oak trees (*Quercus petraea* (Matt.) Liebl.) in a forest in Western Poland. The caterpillars were classified as casebearers (*Coleophora* spp. that mine into leaves from a portable case, made of silk and plant tissue), semi-concealed caterpillars (external feeders that construct shelters by rolling, joining, or bending leaves), and free-living caterpillars (external feeders that do not live inside shelters). We described the abundance and species composition of these insect assemblages and considered the effects of budburst phenology, tree size, genotype, and host trees’ neighbourhood. As genotype, we used distances between samples based on their microsatellite profile, and the genome-wide metrics individual heterozygosity and genome size. The tree’s neighbourhood was characterized as phylogenetic isolation (the average phylogenetic distance to the neighbouring trees), and phylogenetic heterogeneity (the variation in phylogenetic distance to the neighbouring trees). Sessile oaks were chosen because they (a) are important elements in European temperate forests [84], (b) have rich and diverse insect faunas [85,86], (c) are genetically diverse (including hybridization and introgression between *Q. petraea* and *Q. robur* L., [84,87,88,89]), and (d) provide ready comparisons with a rich literature.

## 2. Material and Methods

### 2.1. Study Area of Focal Trees

We selected 25 individual sessile oak trees (*Q. petraea*) that form part of the canopy of the Puszcza Zielonka forest in Western Poland (52°33′12″ N 17°06′48″ E). The Puszcza Zielonka covers 12,202 ha of mostly managed forest, consisting of mainly sessile oaks and pines (*Pinus sylvestris* L.), often in mixed stands [90]. In some areas, there is a sub-canopy of common hornbeam (*Carpinus betulus* L.). Near well-used roads, there is often an understorey consisting mainly of common hazel (*Corylus avellana* L.), the invasive black cherry (*Prunus serotina* Ehrh.), or bird cherry (*Prunus padus* L.). However, in most places, the understorey is kept short by grazing wildlife. Other common trees include European beech (*Fagus sylvatica* L.), black alder (*Alnus glutinosa* Gaertn.), pedunculate oak (*Q. robur*), spruce [*Picea abies* (L.) H. Kars], larch (*Larix occidentalis* Nutt.), silver birch (*Betula pendula* Roth.), maple (*Acer pseudoplatanus* L., *A. platanoides* L.), and elm (Ulmus glabra Huds; [90]).

We chose trees in two clusters (near the villages Zielonka and Kamińsko), and within clusters in groups of two or three trees that differed in neighbourhood. The minimum distance between two chosen trees was 20 meters. The maximum distance between groups was 7.4 kilometres. Because *Q. petraea* predominantly occurs on dryer soils, focal oak neighbourhoods consisted primarily of other oaks, pine trees, beech trees, and hornbeams. Thus, while the main tree species are similar to those in previous studies on effects of phylogenetic isolation, the diversity of trees was lower [40]. In addition, we focused on sessile oaks rather than including both *Q. petraea* and *Q. robur*.

### 2.2. Phylogenetic Neighbourhood of Focal Trees

To calculate neighbourhood characteristics of focal trees, we used phylogenetic crown ages following Vialatte et al. [40]. The original publications used were [91,92,93,94,95]. Crown age represents the time when the oak lineage and the other lineage started to be physiologically and physically distinct from the perspective of phytophagous insects. This prevents the data from being dominated by the difference between angiosperms and gymnosperms. We omitted the understorey (not in touch with the crown of focal trees) in the characterization of the neighbourhood of individual trees because the understorey is not a major source of colonists for the canopy [96,97]. We calculated the degree of phylogenetic isolation as the average phylogenetic distance to neighbouring trees [40], and phylogenetic heterogeneity as the standard deviation of phylogenetic isolation (new metric).

### 2.3. Budburst Phenology and Tree Size

To estimate the day of 50% budburst for each focal tree, we visited all focal oak trees every three to four days during the period of oak budburst in 2019 (15 April–14 May). One observer scored four sides of the crown of each focal tree as having 1, 5, 10, 15, 10, 20, 30, etc. per cent budburst using binoculars. Notably, even at what we call “100% budburst”, oaks may have many resting buds. We then calculated the average per cent budburst for each tree for each observation day, and plotted the percentages against time of observation for each tree. We interpolated the day of 50% budburst from these plots. We measured the circumference at breast height for all the trees in the spring of 2019, and calculated trunk diameter (DBH) assuming a circular cross-section.

### 2.4. Genetic Characterization of Oaks

To evaluate the genetic population structure and diversity of oaks, we performed a population genetic investigation on 51 trees, considering between 17 SSR and 12 SSR loci depending on the analysis. For each tree, DNA was extracted from a small part of a winter bud (Appendix A). To cytologically characterize individual oak trees, we evaluated nuclear DNA content using young leaves, and propidium iodide as DNA stains (internal standard, *Raphanus sativus* “Saxa”. For 49 trees, we measured the DNA content of 10 adult leaves (Appendix A).

### 2.5. Caterpillar Collection and Rearing

We sampled the caterpillar communities of the focal trees between 17 May and 3 June 2019. From each tree, we took 2–4 branches by shooting a rope over a branch using a slingshot and either tearing the branch off [98], or sawing it off using the Short Cut Chain Saw with Guidex (Sodiel International, Lyon, France) or a dehorning wire connected to a rope. We caught the branches on a blue sheet and placed any caterpillars found on the sheet individually into vials with a piece of oak leaf. We then cut each branch into pieces and transported them to the laboratory in plastic bags. The next day, we carefully searched the branches for caterpillars, and placed each individual caterpillar in a 50 mL vial with a piece of oak leaf. We categorized caterpillars as casebearers, semi-concealed, or free-living. Free-living caterpillars were identified to family. We dried and weighed all the leaves to quantify the sampling effort for each branch. We reared the caterpillars to adulthood on leaves of *Q. petraea* at room temperature. We set and dried eclosing moths, and preserved parasitoids in 70% ethanol. We identified moths using taxonomic literature [99,100,101] and the Lepiforum [102] website. Since *Coleophora lutipennella* (Zell.) and *C. flavipennella* (Dup.) cannot be distinguished by the shape of their cases or external features of adults, we identified them based on the morphology of genitalia [103]. Other *Coleophora* spp. could be identified based on the characteristic cases [104], even when they did not produce adults. For each species, we derived information on wingspan (as a proxy of dispersal capacity; [105,106,107]) and host-plant use from the websites Plant Parasites of Europe [104], Catalogue of the Lepidoptera of Belgium [108], and Lepidoptera Mundi [109]. We calculated the community weighted average wingspan, and the proportion of individuals that was classified as host-plant specialist (proportion specialists). In addition, we classified species that only feed on oak out of the main neighbouring trees in this study (oak, beech, hornbeam, pine) as particular specialists and calculated the proportion of such individuals for each tree (proportion of particular specialists).

### 2.6. Data Analysis 

We studied the total number of caterpillars collected, as well as the abundance of the guilds casebearers, semiconcealed, and free-living caterpillars, the three most common species, and the most common free-living caterpillar family. The analyses at the species level were necessarily limited to caterpillars that were reared to adulthood. For each group, the density of caterpillars was calculated as the number of caterpillars divided by the total dry mass of the leaves sampled. To obtain normally distributed residuals, caterpillar density was square root transformed for statistical analyses. We first fitted mixed models for total caterpillar density at the branch level with tree identity as a random effect using the R package ‘lme4’ [110,111] and including branch height as a factor. This revealed that one tree had exceptionally high caterpillar densities on a low sun-exposed branch. Such branches were not sampled from other trees. When this outlier was removed from the data, branch position did not predict caterpillar abundance (results not shown). We then excluded this branch and pooled data from multiple branches per tree for the final analyses. We calculated the Simpson diversity (1-D) based on both the moths reared from each tree and identified cases of casebearers using the R package ‘vegan’ [111,112]. We calculated the parasitism rate for each tree as the proportion of parasitoids.

To test the predictions in Figure 1 regarding abundance, diversity, functional traits, and parasitism, we used a combination of OLS linear models (for linear predictors) and Mantel tests [110,113]. To perform model selection, we took the “dredge” approach using the R Package ‘MuMIn’ [114]. Thus, for each dependent variable, this function fitted all possible OLS models with the predictors and their interactions with phylogenetic isolation, and then sorted the results by the value of the Akaike Information Criterium corrected for small sample size (AICc). We subsequently took the ten best-fitting models (Appendix A), and determined for each predictor and interaction how often it was included in these ten models (Table 1). We also noted the direction of the effect if it was consistent among models and was not part of an interaction (Table 1). To gauge how these ten best-fitting models performed differently from each other, we report the difference in AICc value between the top model and the tenth best-fitting model (Table 1). We examined the results of the top model and performed outlier exclusion when warranted (Appendix A), and then indicated the level of significance of the selected predictors in the top model (Table 1). To test how the similarity among trees in caterpillar abundance is predicted by the genetic divergence among trees, we performed Mantel tests [113]. We used Provesti distances calculated from microsatellite data (Appendix A) relative to similarity in the square root of caterpillar density and similarity in upper model residuals, using the R package ‘ade4’ [115]. To estimate which variables predict community composition, we performed a Permutational Multivariate Analysis of Variance (PERMANOVA) using the R package ‘vegan’ [111,112,116], which partitions the variation in community composition to potential predictors [117]. 

## 3. Results

### 3.1. Overview of the Data

As a result of our deliberate selection of focal trees, the degree of phylogenetic isolation ranged from zero (three trees surrounded by oaks) to 140 million years (three trees surrounded by pines or spruce). Among trees with intermediate phylogenetic isolation, the phylogenetic heterogeneity (standard deviation of phylogenetic isolation) ranged from zero (neighbours are all either beech or hornbeam) to 76.7 (neighbours are a mix of oaks and pines; Appendix A). The budburst date of the focal trees ranged over 16 days, and DBH ranged from 40.1 to 65.9 cm (Appendix A). We found significant differences between trees in microsatellite characteristics, individual heterozygosity, and genome size, and only minor spatial structure differences in the microsatellite data (Appendix A). We collected 612 caterpillars from 25 focal trees, of which 179 were adult Lepidoptera and 126 produced parasitoids. We identified 214 individuals belonging to 22 species (combining adult Lepidoptera and casebearer cases). The three most common species were two casebearers—*C. lutipennella* and *C. flavipennella*—and one semi-concealed species—*Carcina quercana* (Fab.)—and geometrids were the most common family of free-living caterpillars (Appendix A).

### 3.2. Predictors of Caterpillar Abundance, Parasitism, Simpson Diversity, and Functional Traits

We explored how insect community parameters were affected by tree phenotypic and genetic traits and neighbourhood characteristics (Figure 1). In each case, the model comparison based on AICc did not yield a clearly favoured model (Appendix A): the difference in AICc between the top model and the tenth best-fitting model did not exceed 3.3 (Table 1). Therefore, it was warranted to take into account the frequency of occurrence of predictors in the ten best-fitting models, rather than to focus only on the top model. Caterpillar abundance was mainly affected by the phylogenetic neighbourhood of the tree (all major guilds), and by parasitism rate (all three common species and Geometridae; Table 1). The effect of phylogenetic isolation was always positive: more phylogenetically isolated trees tended to have higher caterpillar abundance (Figure 2). On trees with higher parasitism rates, we tended to find lower abundances of co-dominant species (Table 1, Figure 3). Since this effect remained when reconstructing abundances before mortality due to parasitism (based on overall parasitism rates per tree; Appendix A), this is unlikely to be an artefact. No interaction between phylogenetic isolation and other predictors was selected in the top models.

On trees that were sampled later during the 16-day sampling period, we tended to find far fewer free-living caterpillars, but the timing of sampling hardly affected the number of casebearers and semi-concealed caterpillars (Table 1). Budburst date and tree size had no significant effects on caterpillar abundance (Table 1).

Casebearers were less abundant on trees with higher individual heterozygosity (eight out of ten models including the top model; Table 1, Figure 2). One common casebearer species, *C. lutipennella*, appeared to respond to genetic divergence (the results of the Mantel test using raw abundance was significant, but not on residuals of the top model; Table 1). Individual heterozygosity did not have significant effects on the abundance of semi-concealed and free-living caterpillars. Genome size did not predict any of the dependent variables considered.

The parasitism rate was higher on larger trees, and lower in phylogenetically homogeneously distant neighbourhoods (Table 1, Figure 4). On large trees, the Simpson diversity of caterpillars was lower when they were more phylogenetically isolated, but there was no effect of phylogenetic isolation on smaller trees (Table 1, Figure 5). Simpson diversity was lower on trees with higher individual heterozygosity (Table 1), but this was only significant when the interaction between trunk diameter and phylogenetic isolation was included in the model (see also Figure 5). Therefore, this result is not robust albeit being significant in the top model (Appendix A). The proportion of specialists tended to be lower when the parasitism rate was higher (Table 1). Community weighted average wingspan and the proportion of particular specialists were not significantly affected by any of the predictors (Table 1).

### 3.3. Predictors of Caterpillar Community Composition

Caterpillar community composition (identified adults and casebearer cases) was mainly affected by sampling date (Table 2), reflecting that trees that were sampled later yielded lower numbers of free-living caterpillars, while semi-concealed caterpillars and casebearers remained at similar abundances (Table 1). Caterpillar community composition was also significantly predicted by the level of individual heterozygosity of the tree, and by both phylogenetic isolation and phylogenetic heterogeneity. Parasitism rate was a marginally significant predictor of caterpillar community composition (Table 2), probably reflecting that co-dominant species were more affected by parasitism than rarer species (Table 1). 

## 4. Discussion

To estimate the relative contribution of developmental, micro, and macroevolutionary processes to determining insect community parameters on individual trees, we sampled the caterpillar faunas of individual oak trees. In order of importance and robustness, our main findings are as follows. First, the dominant effect was that the presence of phylogenetically distant trees in the neighbourhood of a focal tree increased caterpillar abundance. Moreover, phylogenetic isolation and heterogeneity of the neighbourhood allowed us to predict the caterpillar community’s composition reasonably. Furthermore, among larger trees, caterpillar communities tended to be less diverse when neighbouring trees were more distantly related. However, the effects of phylogenetic isolation on community composition and diversity were not accompanied by shifts in community averaged wingspan or the proportion of host-plant specialists. Second, the other consistent effect was that higher parasitism rates were associated with lower abundances of the most common caterpillar taxa. Third, we found that free-living caterpillars were only abundant for a short period after budburst, while the abundance of casebearers and semi-concealed caterpillars did not decline during the study period. Fourth, our data indicate that individual heterozygosity of trees affected caterpillar community composition as it reduced the abundance of casebearers. Fifth, caterpillar parasitism rates were higher on larger trees, and lower on those occurring in phylogenetically homogenously distant neighbourhoods. Sixth, we found no evidence that neighbourhood affected local adaptation. In total, 50% of caterpillars died without producing an adult moth or parasitoid, as is typical for this type of study [80]. Molecular methods could be used in future to evaluate the possible effects of parasitism on other causes of caterpillar death, but we have no reason to suspect that such bias would differ between trees. Our results should be substantiated by increasing the number of replicates, with multiple samplings within a season, and replicated across years and regions.

### 4.1. Why Was Caterpillar Abundance Higher on More Phylogenetically Isolated Trees, Opposite to Findings of Most Previous Studies?

The increase of caterpillar abundance that we found on more phylogenetically isolated trees is opposite to the decreases in insect herbivore abundance and leaf damage found in other studies [40,41], and commonly reported decreases in herbivory in more (phylogenetically) diverse vegetations [42,118,119]. However, our results are similar to those for oribatid mite faunas on oak branches [120], and predation by rodents on oak seeds [121]. One of the differences between earlier studies on tree herbivory and ours is that we worked on particularly old trees. Our trees were on average larger (average DBH 51.9 cm) than those used by Vialatte et al. ([40]; 29.7 cm) and Yguel et al. ([41]; 19.8 cm). Thus, we might speculate that different processes dominate at different stages of tree development (similar to [50]). At the seed level, concentration effects (predators and parasites concentrating on the few resources) appear to drive higher attack rates on seeds in more phylogenetically isolated neighbourhoods [121]. For intermediate-sized trees, the difficulty of reaching phylogenetically isolated trees appears to reduce insect abundance and diversity (Figure 1, arrow 4; [40]). On large trees, such effects on insect movements may then be reduced, and other mechanisms may dominate. At least four mechanisms may explain our finding of increased caterpillar abundance on more phylogenetically isolated tees. First, more phylogenetically isolated trees may have higher leaf quality [34,122], which attracts or supports higher insect abundance (Figure 1, arrow 3; [123]). Second, specialized insects may initially disperse randomly, and when they happen to reach a phylogenetically diverse neighbourhood, orient towards oaks, becoming concentrated on the few suitable trees (Figure 1, arrow 4; [120,124,125]). When these insects instead happen to reach a neighbourhood dominated by oaks, they will remain evenly distributed, and thus at low densities on individual trees. Third, insects may have a lower propensity to leave trees in a distantly related neighbourhood, even when these trees are being overexploited, because insects there have less information on nearby alternatives (Figure 1, arrow 4; [125,126]). As a result, isolated trees may, on average, have higher densities of herbivorous insects. Fourth, phylogenetic isolation may reduce parasitism (Figure 1, arrow 7; [51]) or predation pressure, allowing herbivorous insect populations to become larger (Figure 1, arrow 6). In our study, this trend was not significant (parasitism was only present in two out of ten best-fitting models; Table 1). Nevertheless, parasitism could play a role, because higher parasitism rates were associated with lower caterpillar abundances (Table 1) and parasitism rates tended to be high overall for those caterpillars that did not die of other causes (mean 43%).

### 4.2. How Could Neighbourhood Have Affected Community Composition?

Phylogenetic isolation of trees could have affected insect community composition by reducing the colonization rate of specialized species of herbivorous insects (Figure 1, arrow 4; [40]). However, we did not find a lower proportion of specialized individuals on more phylogenetically isolated trees. Phylogenetic isolation could also affect community composition if more widely dispersing species are more likely to colonize more phylogenetically isolated trees [40] or become concentrated on isolated trees [124]. Assuming that species with larger wings are better dispersers [106], we expected to find more individuals of species with larger wingspans on more phylogenetically isolated trees. However, we did not find an effect of phylogenetic isolation on the community weighted average wingspan. Thus, insect colonization and extinction on individual trees appears to be predominantly neutral. Perhaps moth dispersal in mixed forests is not strongly related to host-plant specialization and wingspan [127]. Alternatively, the mechanisms through which tree neighbourhood affects caterpillar community composition may be dominated by effects other than insect movements. In particular, traits of focal trees may be affected by their neighbourhood (Figure 1, arrow 3; [123]).

### 4.3. Why Is the Effect of Phylogenetic Isolation on Caterpillar Abundance Strong for Entire Caterpillar Guilds, but Not for the Dominant Species or Dominant Caterpillar Family?

That phylogenetic isolation increases general caterpillar abundance of, but not of the individual dominant species, suggests that phylogenetic isolation of trees increases the abundance of many rarer caterpillar species. This difference in responses to phylogenetic isolation of rare vs. common species may be due to differences in dispersal strategy. In particular, dominant species might have large resident populations on individual trees, while less abundant species more often disperse widely and then search for suitable hosts (known as ‘in situ reproductive recruitment vs. immigration’; [128]). The resident populations will then not be affected by phylogenetic isolation, while widely dispersing species would often show concentration effects in phylogenetically diverse neighbourhoods [120,124].

### 4.4. Why Did Only Casebearers Respond to Genetic Traits?

Casebearers had a lower abundance on trees with higher individual heterozygosity. We also found a correlation between genetic divergence and similarity in abundance in the casebearer *C. lutipennella* (Figure 1, arrow 2). Although genotype–phenotype (e.g., leaf quality) associations are often complex and difficult to disentangle, we speculate that casebearers may be particularly sensitive to increases in the diversity of leaf defences that may be caused by high levels of heterozygosity. This is only speculative because we do not know whether the level of heterozygosity we discovered represents a local or a genome-wide effect. Moreover, there are no data on whether the microsatellite markers are linked to any genes responsible for leaf traits that are essential to herbivores. The absence of significant relationships between heterozygosity and abundance of other guilds might reflect that under field conditions, any effect of genetic differences between trees is less important than effects of the environment [22,23,24]. This difference in response to heterozygosity between guilds is unexpected, because leaf miners (the casebearers in our study) are generally believed to be better at avoiding plant defences than external feeders [73]. However, successful manipulation of leaves might require particular combinations of host and leaf miner genes [59,129]. Perhaps no effects of genetic traits on the abundance or diversity of other herbivorous insects could be detected because insects have responded to host population structure by evolving lineages that are adapted to the various tree genotypes [59,60,61]. Alternatively, most herbivorous insects could be generalists with respect to the within-species lineages present in a vegetation. The effect of individual heterozygosity on Simpson diversity was not robust.

### 4.5. Why Is the Effect of Parasitism on Caterpillar Abundance Stronger for the Dominant Species Than for All Species Combined?

That parasitism decreases the abundance of common species but not others can be explained by functional responses to host density in parasitoids and by differences in dispersal strategies between dominant and rarer herbivore species. Since parasitoids usually concentrate their activity in patches with high host density [130,131,132], parasitoids can be expected to affect common caterpillar species more than rare ones. Furthermore, suppose that rarer herbivore species indeed disperse more widely than common species (as argued above). In that case, at a given tree and for a given caterpillar species, the parasitism rate during the previous year will have little effect on this year’s abundance. In contrast, resident caterpillar populations would exhibit host-parasitoid population dynamics at the scale of individual trees, in case parasitoids also form resident populations. Across trees, only these resident populations would show a significant relationship between caterpillar abundance and parasitism.

### 4.6. Why Was Parasitism Rate High on Large Trees?

Parasitism rate was higher on larger trees, corroborating Klapwijk and Björkman ([133]; Figure 1, arrow 8). Perhaps larger trees are easier to find for parasitoids because they are more visually apparent, harbor a larger number of caterpillars, or emit stronger chemical signals [44,134]. Larger trees could also support larger parasitoid populations with lower extinction risks [53,135], leading to higher parasitism rates. In addition, larger trees are likely to be older, increasing the probability of being colonized by parasitoid species that are capable of using this particular host tree and its particular herbivores. Overall, parasitoids appeared to play an important role in determining the abundances of caterpillar species among trees. Similarly, other natural enemies may have been important in determining caterpillar communities on our study trees [55].

### 4.7. Why Does the Abundance of Free-Living Caterpillars Rapidly Decrease over Time?

We found that later sampling dates yielded fewer free-living caterpillars, while other guilds were not affected by the sampling date. Spring caterpillars typically perform best on soft buds and very young leaves that may be less defended than older leaves [1,2]. Such food may allow high growth rates and thus a short development time. Free-living caterpillars are especially selected to develop fast to avoid predation, while semi-concealed caterpillars tend to experience lower daily predation rates [79] and can thus tolerate slower development. Such differences in growth rate and dependence on young leaves between free-living and semi-concealed caterpillars could explain the rapid decrease in free-living caterpillar density, which was not observed in semi-concealed caterpillars. Casebearers do not figure in this discussion because their cases can remain on the leaves when they are parasitized or have pupated.

## 5. Conclusions

Albeit only correlative, our results suggest that leaf development across days decreases free-living caterpillar abundance; tree growth across decades increases parasitism of herbivores; heterozygosity in genetic characters evolved across millennia decreases casebearer abundance; high phylogenetic distance of oaks from their neighbours over millions of years increases the abundance of entire caterpillar guilds, decreases herbivore diversity, and reduces parasitism when neighbours are consistently distantly related; and parasitism in turn decreases the abundance of dominant caterpillar species. Our results further suggest that these relationships are driven by local trophic interactions selecting for particular herbivore species rather than by random movements among trees. This mosaic of processes playing out on very different scales might help to maintain the enigmatic diversity of insects on trees. Notably, the increase in caterpillar abundance in distantly related neighbourhoods would disfavour oaks in phylogenetically diverse stands, contrary to results of most previous studies [40,41,42,118]. Therefore, the consequences of growing amongst distantly related tree species may depend on factors such as geographic region and tree age.

## Figures and Tables

**Figure 1 insects-13-00367-f001:**
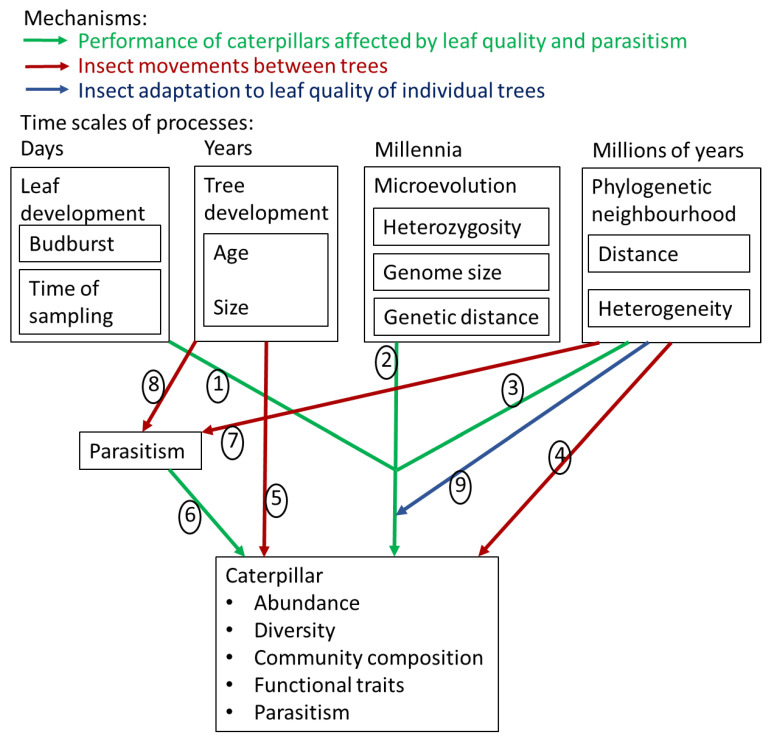
Mechanisms through which processes at different time scales could affect caterpillar communities on individual trees as explained in the Introduction. The importance of particular mechanisms could vary between guilds. Note that both red and green arrows will affect the abundance and diversity of insects. Green arrows (i.e. leaf quality and parasitism) will in addition favour particular species and thereby affect community composition. Red arrows (movement) will either not affect community composition if dispersal is entirely random across taxa, or increase the proportion of good dispersers. The blue arrow depicts interaction terms between phylogenetic distance of the neighbourhood and tree traits.

**Figure 2 insects-13-00367-f002:**
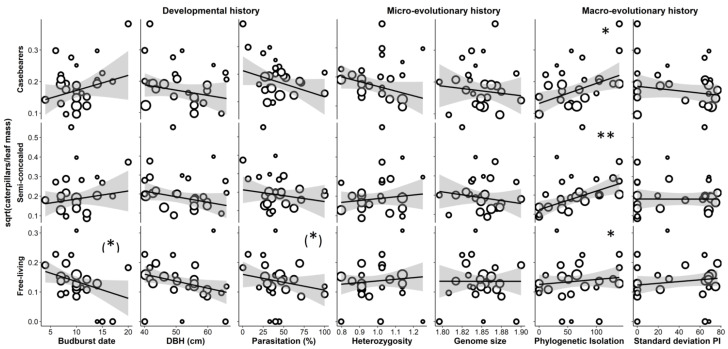
Relationships between caterpillar abundance and tree characteristics per guild weighted by leaf mass. The characteristics of focal trees are ordered by the time scale at which each process plays out. Data derived from larger samples are depicted with larger circles. Relationships that were significant in OLS regressions are indicated with (*) *p* < 0.1, * *p* < 0.05, ** *p* < 0.01 and grey bands depict standard error. Spearman correlation coefficients are given in Appendix A, and results of multiple regression analyses are summarized in Table 1.

**Figure 3 insects-13-00367-f003:**
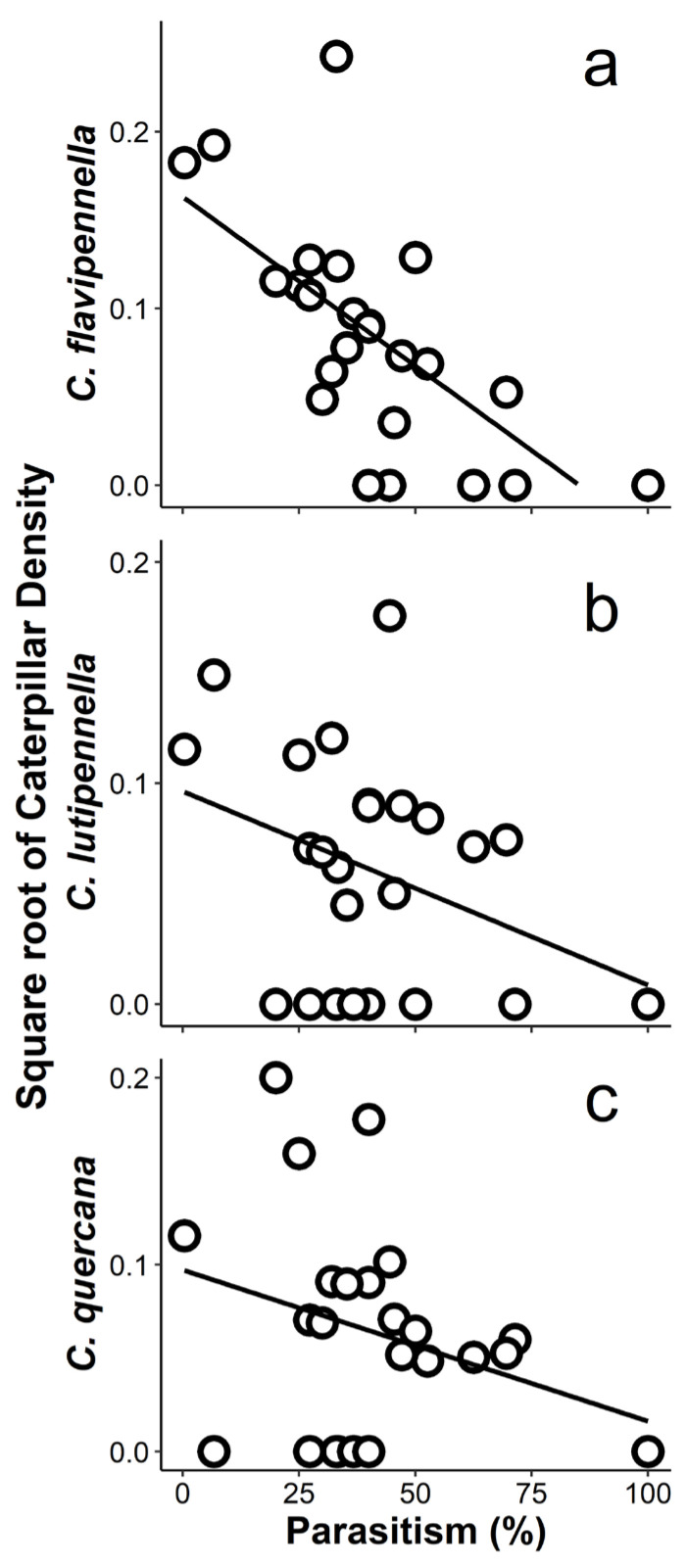
Relationships between the abundance of the three most common caterpillar species and overall parasitism on a tree., with (**a**) *Colephora flavipennella* OLS regression: R^2^ = 49%, *p* < 0.001, (**b**) *Colephora lutipennella* R^2^ = 16%, *p* = 0.051, (**c**) *Carcina quercana* R^2^ = 11%, *p* = 0.103. In multiple regression analyses, the abundance of both *Coleophora* species was significantly predicted by parasitism, and for *C. quercana,* parasitism was selected into the top model but was marginally significant (Table 1, Appendix A). Since a given species may have been absent while parasitism rates were calculated for other species, there are points of zero abundance in these graphs.

**Figure 4 insects-13-00367-f004:**
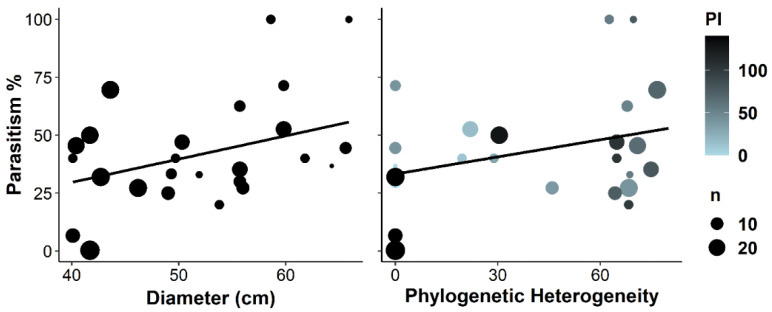
Relationships between caterpillar parasitism rate and trunk diameter, and phylogenetic heterogeneity of the neighbourhood (standard deviation of phylogenetic isolation). PI = phylogenetic isolation, *n* = sample size. In multiple regression analyses, trunk diameter and phylogenetic heterogeneity of the neighbourhood were included in the top model with trunk diameter being significant and phylogenetic heterogeneity marginally significant (Table 1, Appendix A).

**Figure 5 insects-13-00367-f005:**
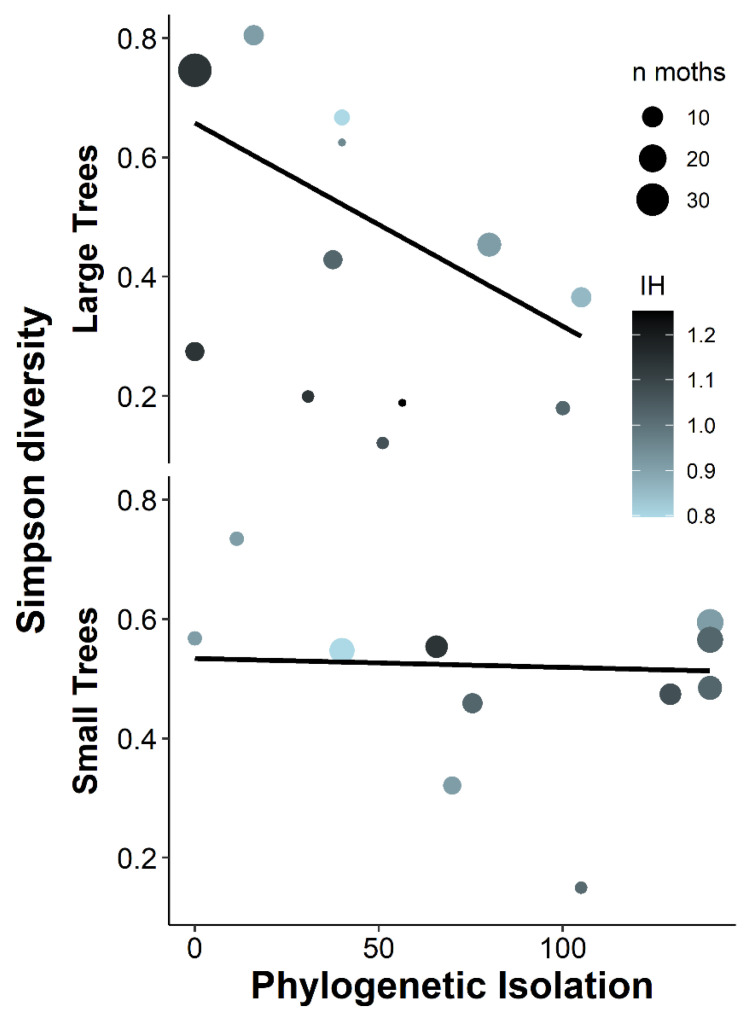
Simpson diversity of caterpillar communities based on larvae reared to adults and casebearer cases from small (DBH < 50 cm) and large (>50 cm) individual trees, illustrating the model with the lowest AICc value (Table 1). OLS Regression lines illustrate the interactive effects of phylogenetic isolation and tree size and use the number of identified moths (n moths) as weight. Both when using a large-small dichotomy and when using all continuous predictors, there is a significant interaction between diameter and phylogenetic isolation (Table 1). The additional effect of heterozygosity (IH, Table 1) is only significant if the interaction between DBH and phylogenetic isolation is accounted for. This probably reflects that for larger trees, points above the regression line tend to have lower heterozygosity than those below it. Note that the range of phylogenetic isolation is narrower for larger trees.

**Table 1 insects-13-00367-t001:** Summary of results for caterpillar abundances, parasitism rate, Simpson diversity, and functional traits. The models used data from 25 trees, except for Simpson diversity and insect traits, where two trees were excluded due to a lack of reared adults, and for Mantel tests, one tree was excluded due to a lack of genetic data. The number of models for which each predictor was selected out of the ten best-fitting models is given for each dependent variable. The direction of the effect is indicated with a plus or minus sign if it is consistent among models and not part of an interaction term. Predictors that were included in the top model are surrounded by black lines, with. *p* < 0.1, * *p* < 0.05, ** *p* < 0.01, and *** *p* < 0.001. The delta range denotes the difference in AICc between the top model and the tenth best-fitting model (Appendix A). *p*-values from Mantel tests between genetic divergence and similarity in caterpillar abundance are given to the right, where ‘Raw’ is based on the raw abundance data, and ‘Residuals’ on the residuals of top models. n = the total number of caterpillars or moths, Cluster = locality, Day = sampling day, BB = date of 50% budburst, DBH = trunk diameter at breast height, Par = parasitism rate, Fcyt = genome size (2C nuclear DNA content (pg)), IH = stand-ardized individual heterozygosity based on the mean observed heterozygosity, PI = phylogenetic isolation (ma), sdPI = phylogenetic heterogeneity of the neighbourhood expressed as the standard deviation of phylogenetic isolation, P. = proportion, Spec. = specialists.

		Number of Linear Models out of Top 10 with:																	Mantel Tests	
		Main Effect:									Interactions with PI							Range	*p*-Values	
Dependent Variable	n	Cluster	Day	BB	DBH	Par	IH	Fcyt	PI	sdPI	Day	BB	Diam	Par	IH	Fcyt	sdPI	Delta	Raw	Residuals
**All caterpillars**	612	1	3−	0	1−	4	0	1+	10+ **	1−	0	0	0	2−	0	0	0	2.8	0.400	0.541
**Casebearers**	214	3	2−	0	0	9−	8−	0	10 *	1−	0	0	0	8−	5+	0	0	3.3	0.153	0.094
*C. flavipennella*	48	0	1−	3−	1−	10− ***	3−	1−	4	0	0	0	0	0	2+	0	0	2.7	0.212	0.230
*C. lutipennella*	37	10	1−	5+	1−	7− ***	2−	2−	1+	0	0	0	0	0	0	0	0	2.3	0.044 *	0.152
**Semi-concealed**	261	1	2−	1+	2+	1−	1+	1+	10+ **	1−	0	0	0	0	0	0	0	2.9	0.436	0.614
*C. quercana*	35	0	1+	3+	1−	5−	0	0	3+	1+	0	0	0	0	0	0	0	2.7	0.130	0.191
**Free living**	136	1	8−	1−	1+	1−	2+	1+	9+ *	0	0	0	0	0	0	0	0	3.1	0.469	0.462
Geometrids	57	3	3−	1−	4−	4− *	0	0	0	8+ *	0	0	0	0	0	0	0	1.7	0.278	0.379
**Parasitism rates**	126	2	1+	2−	6+		1+	1−	2−	9+ *	0	0	0		0	0	1+	2.6	0.342	0.182
**Simpson diversity**	214	9	2+	1+	8 **	2+	9− *	2−	9 ***	4−	2−	0	8− ***	0	1+	2+	0	3.3	0.38	0.767
Wingspan	214	5	3−	1+	1−	1−	1+	0	1+	1+	0	0	0	0	0	0	0	2.4	0.377	0.411
P. Specialists	214	1	1−	1−	1−	4−	2+	1−	0	1+	0	0	0	0	0	0	0	2.1	0.196	0.271
P. Particular Spec.	214	4	2−	0	2−	1−	1−	2−	1−	0	0	0	0	0	0	0	0	2.4	0.619	0.529

**Table 2 insects-13-00367-t002:** Results of PERMANOVA analysis of caterpillar communities on individual oak trees. SS = sum of squares, MS = mean squares, Day = sampling day, BB= date of 50% budburst, DBH = trunk diameter at breast height, Par = proportion of caterpillars parasitized, IH = standardized individual heterozygosity based on the mean observed heterozygosity, Fcyt = 2C nuclear DNA content (pg)), PI = phylogenetic isolation (ma), sdPI = phylogenetic heterogeneity (standard deviation of phylogenetic isolation). * *p* < 0.05, ** *p* < 0.01.

	Day	BB	Diam	Par	IH	Fcyt	PI	sdPI	Residuals	Total
Df	1	1	1	1	1	1	1	1	16	24
SS	0.51	0.027	0.13	0.217	0.352	0.098	0.276	0.26	1.461	3.33
MS	0.51	0.027	0.129	0.217	0.352	0.098	0.276	0.26	0.091	1
Pseudo-F	5.584	0.292	1.418	2.374	3.856	1.073	3.026	2.844	0.439	
R^2^	0.15	0.01	0.04	0.07	0.11	0.03	0.08	0.08		
*p*	0.006 **	0.886	0.210	0.077	0.017 *	0.35	0.039 *	0.047 *		

## Data Availability

The data presented in this study and not included in the appendices are available on request from the corresponding author.

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
