# Peer review of "What Drives Caterpillar Guilds on a Tree: Enemy Pressure, Leaf or Tree Growth, Genetic Traits, or Phylogenetic Neighbourhood?"

_insects, 2022, doi:10.3390/insects13040367_

Round 1

Reviewer 1 Report

This is an interesting study that appears to be well designed an executed. I am not an expert in all of the methods that were used, but those that I am familiar with appear to be used appropriately. I don't have any specific comments as the paper is well-written and the methods and results are detailed and well-documented. The use of a "phylogenetic heterogeneity" measure is an interesting approach, and appears to work in the context of this study. Some of the results are contrary to other publications related to diversity studies, but reasonable explanations are provided. Overall I recommend publication w/o any issues. I did not have time to make individual text edits but I also did not notice any significant problems.

Author Response

We thank the reviewer for the comprehensive and positive review of our manuscript, and we have attempted to address the concern raised.

We have reread the text and made some changes to improve consistency and readability. In particular, we replaced ‘functional group’ with ‘guild’ and everywhere use the term ‘community composition’ rather than mixing it with ‘species composition’. As an example of improving readability, we split the sentence ‘We categorized caterpillars as casebearers, semi-concealed, and free-living, and free-living caterpillars were identified to family’, it becomes ‘We categorized caterpillars as casebearers, semi-concealed, and free-living. Free-living caterpillars were identified to family level.’

Reviewer 2 Report

The study reported is manuscript is a very interesting contribution to our understanding of the importance of different processes - varying largely in time-scale - to the shaping of folivore insect communities on large trees. This work reflects the large advances that have been made since if was first asked what are the drivers of insect diversity found on trees. Many genomic techniques have been developed that allow more focused questions, and also, much phylogenetic work has been done, that now make it possible to examine ecological and evolutionary time processes simultaneously. The authors have combined many techniques and a well designed sampling design to answer the main question, and this work needs to be published, including to emulate similar studies elsewhere.

The weaknesses of the work - number of replicates, limited sampling within a season and no temporal replication across years, are normally an issue, but the authors have been able to temperate the interpretation of their results. It should not deter to publish this work.

I have read the manuscript throughout and in detail and I found no flaw.

It is well written and the results are clearly presented.

There are two minor comments that I think should be addressed :

(1) Can a better case be made about insect movement and population isolation from a genetic point of view ? I am not sure how isolation would be that strong at the spatial scale considered in this study ?

(2) The title of the manuscript mention guilds but the term "community" is largely used throughout. It is not clear if community is used to represent assemblages of species or of guild (e.g. lines 93-96).

Author Response

We thank the reviewer for her/his  careful reading of the manuscript and the constructive remarks. We first address the main concern and proceed by addressing reviewer-specific comments.

We are indeed conscious of the weaknesses of our work and are happy that you are still recommending publication. To admit the weakness of the study, we added a sentence at the end of the first paragraph of the discussion: “Our results should be substantiated by increasing the number of replicates, with multiple samplings within a season, and replicated across years and regions.”

There are two minor comments that I think should be addressed :

(1) Can a better case be made about insect movement and population isolation from a genetic point of view ? I am not sure how isolation would be that strong at the spatial scale considered in this study ?

We suppose that the referee here refers to our argumentation about possible interaction terms between phylogenetic isolation and other tree characteristics. We indeed argue that such isolation might among others affect the local adaptation to other tree characteristics. It can indeed be surprising to find local adaptation at such a small spatial scale. However, such local adaptation has been found as also reported in the papers already cited. We added two general references about dispersal and adaptation on small spatial scales, and two that indicate that local adaptation to oak trees can occur at small spatial scales for two common species of oak-feeding moth. We added in the Introduction ‘Such swamping may be limited enough for local adaptation to occur even when distances between conspecific trees are small compared to maximum dispersal distances of insects, as realized dispersal distances are often much shorter (Lester et al. 2007, Richardson et al. 2014). Furthermore, the rate of immigration can be especially low for species with wingless females or when females first oviposit near their eclosion site before dispersing to colonise trees that are farther away (Schneider 1984, Van Dongen et al. 1998).’

(2) The title of the manuscript mention guilds but the term "community" is largely used throughout. It is not clear if community is used to represent assemblages of species or of guild (e.g. lines 93-96).

We made it more clear that we talk about communities of species. In line 93, we changed to ‘… why herbivorous insect communities are often more species-rich on larger trees.’

In line 93 we changed to ‘We described the abundance and species composition of these insect assemblages…’ In results line 418, we added ‘Caterpillar community composition (identified adults and casebearer cases)…’. The discussion of community composition talks about how different species may be affected, not guilds.